# Plasmacytoid Dendritic Cells, a Novel Target in Myeloid Neoplasms

**DOI:** 10.3390/cancers14143545

**Published:** 2022-07-21

**Authors:** Xavier Roussel, Francine Garnache Ottou, Florian Renosi

**Affiliations:** 1INSERM, EFS BFC, UMR1098 RIGHT, University of Bourgogne Franche-Comté, 25000 Besancon, France; francine.garnache@efs.sante.fr; 2Department of Clinical Hematology, University Hospital of Besançon, 25000 Besançon, France; 3Etablissement Français du Sang Bourgogne Franche-Comté, Laboratoire d’Hématologie et d’Immunologie Régional, 25020 Besançon, France

**Keywords:** plasmacytoid dendritic cells, mature plasmacytoid dendritic cells proliferation, blastic plasmacytoid dendritic cells neoplasm, acute myeloid leukemia, tumor microenvironment

## Abstract

**Simple Summary:**

Plasmacytoid dendritic cells are the main type I interferon producing cells in humans and are able to modulate innate and adaptive immune responses. Tumor infiltration by plasmacytoid dendritic cells is already described and is associated with poor outcomes in cancers. In hematological diseases, Blastic Plasmacytoid Dendritic Cells Neoplasm (blastic counterpart) is well described, but little is known about tumor infiltration by mature plasmacytoid dendritic cells described in Myeloid Neoplasms (myeloid tumor cells and clonal pDC proliferation). This review provides a comprehensive overview of plasmacytoid dendritic cells in Myeloid Neoplasms.

**Abstract:**

Plasmacytoid dendritic cells (pDC) are the main type I interferon producing cells in humans and are able to modulate innate and adaptive immune responses. Tumor infiltration by plasmacytoid dendritic cells is already well described and is associated with poor outcomes in cancers due to the tolerogenic activity of pDC. In hematological diseases, Blastic Plasmacytoid Dendritic Cells Neoplasm (BPDCN), aggressive leukemia derived from pDCs, is well described, but little is known about tumor infiltration by mature pDC described in Myeloid Neoplasms (MN). Recently, mature pDC proliferation (MPDCP) has been described as a differential diagnosis of BPDCN associated with acute myeloid leukemia (pDC-AML), myelodysplastic syndrome (pDC-MDS) and chronic myelomonocytic leukemia (pDC-CMML). Tumor cells are myeloid blasts and/or mature myeloid cells from related myeloid disorders and pDC derived from a clonal proliferation. The poor prognosis associated with MPDCP requires a better understanding of pDC biology, MN oncogenesis and immune response. This review provides a comprehensive overview about the biological aspects of pDCs, the description of pDC proliferation in MN, and an insight into putative therapies in pDC-AML regarding personalized medicine.

## 1. Introduction

Plasmacytoid dendritic cells (pDCs) are the main type I interferon- (IFN) producing cells in humans and play a key role in modulating the innate and adaptive responses of the immune system [1]. Mature pDC proliferation (MPDCP) associated with myeloid neoplasms corresponds to an emerging entity in recent studies, described as a differential diagnosis of Blastic pDC neoplasms (BPDCN), but not listed as a clearly defined entity in the 2017 update of the *WHO Classification of Tumours of Heamatopoietic and Lymphoid Tissues* [2]. As BPDCN, aggressive leukemia derived from pDCs, MPDCP is a rare entity, still insufficiently described, especially when this MPDCP is associated with acute myeloid leukemia (AML). Recently, two teams have better characterized MPDCP in AML, and renamed this new emerging entity pDC-AML [3,4]. By extension, MPDCP associated with myelodysplastic syndrome (MDS) and chronic myelomonocytic leukemia (CMML) can be referred to as pDC-MDS and pDC-CMML. The role of this pDC proliferation in AML is puzzling and not well known. Nevertheless, it has been reported in solid tumors that pDC proliferation in the tumor microenvironment presents an adverse impact, in such a way that pDCs play a role in the regulation of the tumor microenvironment [5]. The aim of this review is to present biological aspects of pDCs that could help to better define pDCs in the tumor microenvironment, particularly in myeloid neoplasm, and to provide a general overview of putative therapies in pDC-AML in regard to personalized medicine.

## 2. Biology and Functions of pDCs

### 2.1. Definition of pDC Subset

Despite the original description of pDCs in the 1950s, the definitive description of pDCs as a unique cell type was published in 1999 as the main IFN-I producing cells in humans [1,6]. Human pDCs were originally defined by a plasmacytoid morphology, the lack of lineage markers (lin^−^) and CD11c, the expression of CD4 and major histocompatibility complex (MHC) class II, and the differential expression of immunoglobulin-like transcript receptors 1 and 3 (ILT1 and ILT3), which initially distinguish pDCs from conventional DCs (cDCs) by a higher ILT3 expression than ILT1 [6,7,8]. In contrast to mouse pDCs, which are CD11c intermediate, human pDCs are CD11c^−^ (Table 1) [1].

More recent studies have characterized human pDCs as lin^−^ (cMPO^−^CD3^−^CD19^−^CD14^−^CD16^−^), CD11c^−^, CD303^+^ (BDCA2 or CLEC4C) and ILT7^+^, CD4^+^, CD68^+^, CD123^+^ (interleukine-3 receptor α-subunit), but also lymphoid markers positive such as terminal deoxynucleotidyl transferase (TdT), interleukine-7 receptor (IL-7R) and recombination activating gene 1 (Rag1) [1,6,9,10]. A subset of human pDCs also express CD2 and CD56 that produce lysozyme, defined as CD2^high^ pDCs, but their functions remain unclear [1].

Contrary to cDCs, which are defined as true professional antigen presenting cells (APC), pDCs are defined as professional IFN-I producing cells. Indeed, pDCs are also able to present antigens but at a lower level than cDCs [1,6]. After microbial or CD40L stimulations, pDCs secrete 200 to 1000 times more IFN-I than other blood cells within 1–3 h [7,8]. Thus, this particular capacity to produce a swift and massive IFN production defined pDCs as “natural IFN-producing cells” [11,12]. In addition to innate antiviral immune response, pDCs are also able to prime both tolerogenic and immunogenic adaptive immune response [1].

### 2.2. Development of pDCs

In the bone marrow (BM), pDCs are continually produced [6]. Myeloid progenitor was initially described as a common dendritic cell progenitor (CDP), which appeared to be able to generate both pDCs and cDCs but no other cell lineages [13]. The CDP is characterized by lin^−^ and by the expression of Fms-like tyrosine kinase 3 (FLT3 or CD135), macrophage colony-stimulating factor receptor (M-CSFR or CSF1R or CD115), and the stem cell factor receptor tyrosine kinase KIT (CD117 or SCFR) [14]. A lymphoid progenitor initially named lymphoid-primed multipotent progenitor (LMPP) has also been described, which is able to generate pDCs specifically and has been characterized by lin^−^KIT^int/low^FLT3^+^ and M-CSFR^−^ and high expression of E2-2 (TCF4), an essential transcription factor for pDC development [15].

In fact, several alternative pDC geneses and developments from myeloid and/or lymphoid progenitors have been described in a recent review [9]. While the myeloid or lymphoid origin, or both, of pDCs is ambiguous, FLT3-FLT3 ligand (FTL3L) interaction is essential and sufficient to drive pDCs and cDCs development and survival without any other signal, suggesting a common developmental pathway or a functional convergence [9,13]. Thus, the commitment to produce pDC and/or cDC subsets may be determined at the early stages of differentiation prior to the emergence of phenotypically defined DC progenitors.

After their maturation in BM, pDCs migrate from blood into lymphoid organs whereas cDCs migrate into peripheral tissues. pDCs emerge as mature cells into the periphery where they remain non-proliferative with a relatively short life (only few days) [16]. Typically, pDCs primarily reside in lymphoid organs and continuously recirculate through lymphoid organs, where pDCs represent around 0.1–0.5% of nucleated cells [6].

This trafficking of pDCs involves the differential expression of multiple adhesion molecules such as CD62L, P-selectin, glycoprotein ligand 1 (PSGL1), β1 and β2 integrin, and chemokines receptors such as CXCR3, CXCR4, CCR2, CCR5, CCR6, CCR7, CCR9, CCR10 [1]. Retention of pDC in the BM niche requires CXCR4 [17], whereas the co-expression of CXCR4 and CCR7 induces the migration of pDCs in the splenic white pulp. The CD62L and CXCR3 expressions after pDCs activation permit pDCs to enter lymph nodes at high endothelial venules [7]. Tumor expressing CXCL12 can also promote pDCs expressing CXCR4 in the tumor microenvironment [18]. CXCR3 and CCR5 drive pDCs into inflamed tissues, CCR2 permits the recruitment of pDCs in inflamed skin, CCR6 and CCR10 facilitate migration into inflamed tonsil epithelium, and CCR9 promotes thymus and gut migration [1].

### 2.3. Transcriptomic Network of pDC Development

As previously reviewed [6,19], during pDC development, FLT3L and IFN-I act synergistically on lymphoid progenitors and induce the upregulation of FLT3 [20,21]. In contrast, FLT3L-driven pDC development is inhibited by granulocyte-macrophage colony-stimulating factor (GM-CSF) via STAT5 signaling activation and IFN-regulatory factor 8 (IRF8) inhibition [22].

DC development though the FLT3L-driven program is controlled by the transcription factor PU.1 [23], whereas the subsequent specification of pDCs requires the E protein transcription factor TCF4 (E2-2) (Figure 1) [24,25]. IRF8 expression is necessary in progenitors to initiate pDC development program [26], while TCF4 expression can amplify its own expression through a bromodomain-containing protein 4 (BRD4)-dependent feedback loop, a bromodomain and extraterminal protein family (BET) [27]. TCF4, MTG16 and BCL11A factors promote jointly pDC development [28,29,30]. E protein inhibitor ID2 block the TCF4-driven program and promote cDC1 development. MTG16 and ZEB2 repressors inhibit the cDC1 ID2-driven program and promote the pDC TCF4-driven program [31,32].

Then, the pDC differentiation program involves SPIB transcription factor, which facilitates the retention of immature pDCs in the BM, whereas RUNX2 facilitates the exit of mature pDCs from BM into the periphery [33,34,35]. The IFN-I-producing capacity of pDCs is dependent on IRF7, IRF5 and IRF8 [36,37,38]. SPIB and NFATC3 factors facilitate IFN-I genes activation through their interaction with IRF7 [33,39], while RUNX2 directly activates IRF7 expression [35]. On the contrary, MYC represses RUNX2 and limits IFN-I production [40]. Furthermore, IKAROS promotes pDC formation by antagonizing TGF-β, which promotes cDCs [41]. Epigenetic regulators CXXC5 and TET2 promote IRF7 expression by maintaining its promoter hypo-methylated in pDCs [42].

### 2.4. Innate Activation of pDC

Innate activation of pDC leading to IFN-I secretion is mediated by pattern recognition receptors (PRRs) [43] that recognize pathogen-associated molecular patterns (PAMPs) unique to microbes [44] and damage-associated molecular patterns (DAMPs) [45]. PRRs are multiple germ-line-encoded receptors such as TLRs, NOD-like receptors (NLRs), retinoic acid-inducible gene I (RIG-I)-like receptors (RLRs), C-type lectin receptor (CLRs), cytosolic DNA sensors (CDCs) and receptor for advanced glycation end products (RAGE) [43].

Endosomal nucleic acid-sensing Toll-like receptors (TLRs) 7 and 9 are two innate sensors highly expressed in pDCs that recognize viruses and nucleic acids [46]. TLR 7 and TLR9 recognize single-stranded RNA and unmethylated CpG motif-containing DNA, respectively [6]. The recognition of nucleic acid induces a swift and massive secretion of IFN-I. In addition to IFN-β and IFN-α secretions (IFN-I), pDCs are able to produce IFN-III (IFN-λ) and other pro-inflammatory cytokines and cytokines including TNF-α, interleukine-6 (IL-6), IL-12, CXC-chemokine ligand 8 (CXCL8), CXCL10, CC-chemokine ligand 3 (CCL3) and CLL4 (Figure 1) [1,47].

This secretion of IFN-I after TLR7/9 activation was induced via myeloid primary response protein 88 (MYD88)-IRF7 pathway and pro-inflammatory cytokines and chemokines via the MYD88-nuclear factor κB (NF-κB) pathway [48]. Activation of NF-κB and mitogen-activated protein kinases (MAPKs) induce TNFα and IL-6 secretion but also the expression of several costimulatory molecules such as CD80, CD86, and CD40, and MHC class II molecules [48]. The level of MHC class II expression on pDCs depends on their activation, which is related to several receptors activated by numerous ligands [1,6]. MHC class II and co-stimulator molecules expressions allow naïve T-cells priming by pDCs [9].

Of note, the most important CDCs are cyclic GMP-AMP synthase (cGAS) and absent in melanoma 2 (AIM2). Those two CDCs are essential to antimicrobial activity and play a role in cancer and inflammatory disease development [49,50]. Cellular death with uncontrolled DNA damage or DNA degradation deficiency in cancers induce an aberrant activation of the cGAS-stimulator of IFN genes (STING) pathway, promoting inflammation-driven ontogenesis [51,52,53]. STING agonists could also promote T-cells apoptosis [54].

### 2.5. Non-Canonical pDCs

Recent single-cell RNA sequencing studies (scRNAseq) and high-dimensional phenotypic mapping by Cytometry by Time of Flight (CyTOF) studies have allowed us to describe new DC subsets, of which AXL^+^ SIGLEC6^+^ DC (AS-DC), also named non-canonical pDCs or transitional DCs (tDCS), which differ from canonical pDC [55,56,57]. The AS-DC subset is considered a transitional entity between pDC and cDC that presents a limited IFN-I secretion but a better antigen presentation than canonical pDC [57]. AS-DCs represent 2–3% of DCs in scRNAseq and cytometry analysis, are CD123^+^CD11c^−/low^, co-express unique markers AXL, SIGLEC1/6 and SIGLEC2/CD22, and exhibit a continuum of pDCs and cDC markers and signatures [55]. Co-expressed cDCs markers in AS-DCs are CD2, CX3CR1, CD33/SIGLEC3, CD5, and SIGLEC1/CD169 both at protein and mRNA levels [55], and AS-DCs co-express pDC markers such as CD123 but also BDCA2/CD303 in a lower level [56]. Concerning AS-DCs development, they are E2-2-dependent and require TCF4 for their development similarly to canonical pDCs, but they also express ID2 as cDCs [1,6]. AS-DCs also express the costimulatory molecule CD86 and HLA-DR and are able to induce CD4^+^ and CD8^+^ T-cell proliferation, while IFN-I secretion remains marginal [55], demonstrating a predominant antigen-presenting function rather than an IFN-I response

## 3. Blastic pDC Neoplasm

BPDCN is a rare and aggressive leukemia derived from pDCs, which is characterized by skin and lymphoid organ involvement [2,58]. Bone marrow infiltration is frequent and blast population is heterogeneous on morphologic analysis. The typical morphology is medium-sized cells with a peripheral round nucleus, sometimes irregular, with fine chromatin and small nucleoli, associated with faint and heterogeneous basophilic cytoplasm, gray areas, and small vacuoles [58]. Nevertheless, some blasts have a lymphoid morphology, and others have an immature blast morphology. Overall survival for patients who receive allogeneic stem cell transplantation or not is 49 and 8 months, respectively [58]. Cytogenetic abnormalities are frequently observed (59.7% of BPDCN), mostly involving deletions. The mutational landscape of BPDCN impacts epigenetics, the RAS pathway, splicing, kinases signaling and tumor control (Figure 1) [59].

In BPDCN, blastic pDCs always express CD4 and CD123, a high level of HLA-DR and cTCL1, and very frequently CD56, CD304, CD303, CD36, CD38 and CD45RA. Levels of CD4, CD56 and CD303 are lower than normal T-cells, NK-cells and pDCs [58]. Myeloid markers can be expressed such as CD33 and CD15, as well as lymphoid markers such as CD7, CD2, CD22 and cCD79a, respectively, from more to less frequent markers. CD2AP, SPIB and IFN-I-dependent molecule MX1 are also expressed by blastic pDCs [60]. TCF4 has been recently reported as a reliable diagnosis marker for BPDCN [61]. BCL6, IRF4, and BCL2 are also expressed by blastic pDCs in opposition to normal pDCs [62]. Blastic pDCs are usually CD34 negative [58,60].

A transcriptomic analysis reported BPDCN heterogeneity of cases with distinct leukemogenesis, leading to a distinct IFN-I secretion [59]. Overexpression of pDC lineage genes were observed such as *LAMP5* (*BADLAMP*), *HES6*, *LILRB4* (*ILT3*), *LILRA4* (*ILT7*), *IL3RA* (*CD123*), *CLEC4C* (*CD303*), *GLUL*, *IRF7*, *TLR7*, and HLA class II genes (Figure 1). Moreover, the overexpression of *RUNX2* and *FLT3*, as well as aberrant activation of the NF-κB pathway, has previously been reported by gene-expression profiling and array-based CGH [63,64]. Approximatively 40% of BPDCN express various markers, such as SIGLEC6 and CD22, suggesting an AS-DC origin, but the remaining BPDCN cases express genes upregulated in canonical pDCs [59]. Consequently, a clear cell-of-origin for BPDCN between AS-DC and canonical pDC is still uncertain [55,59].

Concerning BPDCN treatment, chemotherapy without allogenic stem cell transplantation is insufficient to obtain a durable response [58]. New strategies are investigated such as proteasome inhibitors [65], anti-apoptotic inhibitors [66] or anti-CD123 therapies [67]. Combination strategies with CD123 therapies has been recently reviewed [68]. Bromodomain and extra-terminal domain inhibitors (BETi) are also able to induce blastic pDCs apoptosis [61].

## 4. Tumor-Infiltrating pDCs in Solid Cancers

Solid cancers studies have highlighted that tumor infiltration by pDCs is associated with an adverse prognosis, which are tolerogenic rather than immunogenic [1,69]. Tumor-infiltrating pDCs present an activated CD40^+^, HLA-DR^+^, CD86^+^ phenotype, as reported in breast cancers [70], and a mature CD4^+^CD303^+^CD123^+^ phenotype as reported in ovarian cancer [71] and melanoma [72].

However, pDCs could also promote an immunogenic response in tumors through TLRs activation and Th17 differentiation [73]. Recently, a study in head and neck squamous cell carcinoma reported the anti-tumor activity of tumor infiltrative-pDCs, especially a subset of activated OX40^+^ pDC [74]. This OX40^+^ pDC subset presents an activated and mature phenotype with an increased level of expression of CD40, CD80, CD86, OX40L, SIGLEC6, AXL, CD25 and 4-1BB [74]. A unique transcriptome profile of OX40^+^ pDC was identified by non-canonical NF-κB signaling, and IFN-I signaling pathways signature [74]. Non-canonical NF-κB signaling stimulation through TNF family member ligands, as OX40L leads to the slow but persistent activation of the NF-κB pathway [75] and is required for DCs cross-priming of CD8^+^ T-cells [76]. In addition, it was also reported in mice that pDCs require cDC1s to achieve cross-priming of potent tumor-associated antigen (TAA)-specific CD8^+^ T-cells by transferring antigens to cDCs through exosomes [77], suggesting a close cooperation between tumor-infiltrating pDCs and cDCs.

Indoleamine 2,3-dioxygenase 1 (IDO1) and inducible T-cell co-stimulator ligand (ICOSL) are two main tolerogenic mechanisms used by unstimulated and alternatively activated pDCs [78,79]. IDO1 and ICOSL expressions by pDCs are correlated with worse outcomes in patients with cancer [70,79,80,81] and induce the differentiation of naïve CD4^+^ T-cell into IL-10-producing regulatory T-cells (Tregs) [78,82,83]. Those tumor-infiltrating pDCs are poor IFN-I producers, because of cytokine dysregulation by transforming growth factor β (TGF-β) and TNF-α secretions in the tumor microenvironment that inhibit the IFN-I secretion of pDCs through the inhibition of IRF-7 expression [84], and favors Treg expansion [1,80,81]. Tregs express the inhibitory signal of immune response cytotoxic T-lymphocyte-associated protein 4 (CTLA-4) and the glucocorticoid-induced tumor necrosis factor receptor family-related receptor (GITR), reflecting a prolonged stimulation by pDCs [69,70,85]. Tumor-infiltrating pDCs are also able to recruit other suppressive cells in the tumor microenvironment such as myeloid suppressive cells [86], IL10^+^CCR7^+^CD8^+^Foxp3^+^ induced Tregs [87,88], and IL-10^+^ IL-17^+^ Foxp3^neg^ type-1 regulatory T cells (Tr1) [89].

In the past few years, several studies have investigated how to modulate and reshape the ability of pDCs and cDCs to promote an antitumor response through tumor-infiltrating lymphocytes (TIL) activation and differentiation [90]. In cancers, pDCs can prime T-cells via many signaling axes from costimulatory molecule interactions (Figure 1) [91]. Targeting those immune checkpoint molecules is a common anti-tumor strategy currently used in cancers. Other strategies are also proposed as TNF superfamily modulation [92], vaccination and adoptive DC transfer therapy [93,94], and IDO1 blockade [82,83] with or without immune checkpoint molecule inhibitors association [95,96,97]. Finally, targeting CD8^+^ regulatory T-cells has also been proposed [98].

## 5. Mature pDCs Proliferation in Myeloid Neoplasms

A description of pDCs in hematological tumors, apart from BPDCN which is their tumor counterpart, is limited in comparison to solid cancers. Mature pDC proliferation (MPDCP) associated with a myeloid neoplasm corresponds to an emerging entity in recent studies, described as a differential diagnosis of BPDCN, but not listed as a clearly defined entity in the 2017 update of the *WHO Classification of Tumours of Heamatopoietic and Lymphoid Tissues* [2]. MPDCP is rare, but not systematically screened in routine hospital practice. It was reported in association with CMML, AML or MDS; more recently it has been referred to as pDC-CMML [99], pDC-AML [3,4] and pDC-MDS [100], respectively. The ontogeny of pDCs is unclear [9], but a clonal maturation from blasts to pDCs in AML was suggested [101,102]. However, tumor infiltration by pDCs from the bone marrow microenvironment in hematological disease could be also suggested [100], as reported in solid cancers. Functions of pDCs in those entities may differ between a blastic maturation into pDCs and a mature pDC infiltration in the tumor microenvironment.

### 5.1. Mature pDCs Proliferation in CMML

CMML is characterized by sustained monocytosis and an overlap of myeloproliferative neoplasms (MPN) and myelodysplastic syndromes (MDS) resulting from a cumulative mutational landscape including *TET2* (60%), *SRSF2* (50%), *ASXL1* (40%), and the oncogenic *RAS* pathway (30%) mutations [103]. Median overall survival of CMML patients range between 16 and 97 months depending on several scores, such as the Mayo Molecular Model score, and between 15 and 30% of patients die from a leukemic transformation in AML. *ASXL1*, *NRAS*, *SETBP1*, and *RUNX1* mutations are independently associated with inferior overall survival [104]. There is usually an increased monocytic population in the classical monocytes fraction (CD14^+^/CD16^−^) which also express other monocytic markers (CCR2, CD36, HLA-DR, and CD11c) and a low level of CX3CR1 [103,105].

Up to 20% of CMML patients also harbor a concurrent inflammatory and autoimmune condition [105]. The expression of Program Death ligand 1 (PD-L1) by CD34^+^ CMML cells was reported higher than in AML and MDS and increased during hypomethylating agent treatment associated with poorer survival, suggesting an immune escape mechanism [106]. Immunosuppression by myeloid-derived suppressive cells has also been suggested in CMML [107] Moreover, DC nodules, composed mainly of pDCs, have long been known to exist in the BM microenvironment in above 20% of CMML patients [108,109], but their functions are not really known [110]. Recently, a study highlighted the clonal maturation of pDCs in above 30% of CMML patients, composed of mature pDC nodules with the lin^−^CD45^+^CD123^+^CD11c^−^CD33^−^HLA-DR^+^CD303^+^CD304^+^ phenotype, often associated with *RAS* pathway mutations and inferior leukemia-free survival (LFS) [99]. More recently, other mature pDC markers have also been reported: CD2AP^+^, BCL11A^+^, TCL-1^+^ but at a lower level than BPDCN [111].

In the original description of pDC-CMML (pDC-rich CMML) from two tertiary centers, the cut-off of pDCs was defined as a least 1.2% of BM mononuclear cells (BMMC) and 0.6% of peripheral blood mononuclear cells [99]. The median pDC rate in pDC-CMML was higher in BMMC than PBMC (0.32% [0.04–0.81] versus 0.10% [0.02–0.26]; *p* < 0.0001). In comparison, pDC-poor CMML (poor rate of pDCs considered as no MPDCP association) present a lower rate of pDCs than healthy donors. A small fraction of pre-DC/AS-DC was also observed in healthy donors, pDC-poor and pDC-rich CMML samples (3.1%, 4.1%, and 0.6%, respectively, but without reaching significant difference). After TLR7 and TLR9 agonist stimulation, pDCs from pDC-CMML produced IFNα secretion. In addition, CD4^+^CD25^high^CD127^low^ Treg rate in BMMC and PBMC was higher in pDC-CMML than healthy donors. RNA-sequencing analysis showed that pDCs from pDC-CMML expressed a similar signature compared to healthy mature pDCs including *HLA-DR*, *CD123*, *CLEC4C* (*CD303*), *TLR9*, *TLR7*, *NRP1* (*CD304*), *IRF7*, *LILRA4* (*ILT7*) and *TCF4* (*E2.2*) genes. Furthermore, pDCs from pDC-CMML expressed a low level of *CD5*, *CD2* and *SIGLEC6* genes usually expressed in pre-DC/AS-DC entities [55,99]. At last, genes implicated in IFN-I signaling pathway and response to IFN-I seemed to be upregulated in pDCs from pDC-CMML in comparison to pDCs from pDC-poor CMML, but do not reach significant difference. This pDC proliferation was also associated with a higher cumulative incidence risk of leukemic transformation (HR 2.59 [1.21–5.51]; *p* = 0.014) [99]. Additionally, CD34^+^ cells from pDC-CMML were hypersensitive to FLT3L in in vitro culture model and FLT3L level was lower in pDC-CMML than pDC-poor CMML.

Finally, more recently, pDC-CMML have been described to be associated with systemic immune dysregulations in a smaller cohort [111]. In DC nodules from BM, IDO1 co-expression was detected on CD11c^+^CD123^dim^ monocyte derived-DCs rather than CD11c^−^CD123^+^ pDCs, but IDO1 co-expression could also be detected on monocytes [111]. Those IDO1 co-expression data reflect complex intercellular interactions and suggest IDO1-expressing myeloid-derived suppressor cells involvement. Moreover, several T-cell subsets were reported in a significant lower quantitative level in CMML with IDO1^+^ DCs in comparison to healthy donor controls, including naïve T-cells, CD8^+^ central memory T-cells, Th1 cells Th1/Th2 ratio, CD4^+^ terminal effector cells, and γδ T cells, whereas CD4^+^ central memory cells and NK cells were expanded [111]. In comparison to CMML without IDO1^+^ DCs, CMML with IDO1^+^ DCs also presented a significantly higher Treg level, and RNA-sequencing confirmed the Treg-associated gene upregulation in CMML with IDO1^+^ DCs [111].

Further in vivo studies are necessary to confirm these data and to propose specific therapies. However, in a recent study using a mouse model, the cooperative role of *NRAS* and *ASXL1* mutations in reprogramming BM immune microenvironment by the AP-1 transcription factor was highlighted [112]. These mutations also led to the upregulation of PD-L1 and CD86, which could enable clonal evolution and disease progression in AML [112,113]. Host-derived T-cells overexpressed PD-1 and T-cell immunoreceptor with immunoglobulin and ITIM domains (TIGIT) receptors, leading to an exhausted phenotype [112]. At the moment, as recently reviewed, several specific targeted drugs are being investigated such as CD123-directed diphtheria toxin tagraxofusp-erzs, anti-JAK inhibitor ruxolitinib, anti-PD-L1 atezolizumab and DNA methyltransferase inhibitor guadecitabine, or anti-IL-1B canakinumab [113]. In addition, the combination of MAP/ERK kinase (MEK) and BET inhibitors could induce the downregulation of FLT3 and AP-1 expression, partial restoration of the immune microenvironment, enhancement of CD8^+^ T-cells cytotoxicity, and prolonged survival in mice [112].

### 5.2. Mature pDCs Proliferation in MDS

Myelodysplastic Syndrome is a group of heterogeneous clonal hematologic malignancies [114]. MDS are characterized by defective BM hematopoiesis and by the occurrence of intramedullary apoptosis. The unbalance between cell death and proliferation in BM has an important role in the pathogenesis of MDS. As described in CMML, MDS could be associated with inflammatory and autoimmune disorders in 10–30% of MDS [115]. As reviewed, abnormal levels of cytokines, chemokines and growth factors in peripheral blood and BM were found in MDS [115]. In MDS, levels of pro-inflammatory cytokines TNFα, TGFβ, IL-6, IL-1 are increased, suggesting a BM inflammatory dysregulation. A higher level of TNFα is associated with a poorer prognosis and leukemic transformation risk [116], and a higher occurrence of apoptosis [117]. However, low levels of IL-10 and IL-4 are associated with better clinical outcomes in MDS [118]. IL-10 is more intensely secreted in high-risk MDS, whereas Th17 T-cells are markedly increased in low risk MDS [119]. Moreover, the Th17/Treg ratio is higher in low risk MDS, whereas it is lower in high risk MDS, and Tregs inhibit IL-17 production. The proportion of CD8^+^ T-cells expressing PD-1 and PD-L1 is greater in high risk MDS [106]. IDO1 level is also increased in MDS irrespective of the risk score reflecting inhibitory effects [120].

Gene expression profiling has highlighted that many signal transducers in the TLR pathway were overexpressed in 40–80% of MDS patients [115]. Overexpression of *TLR4* was reported in BMMC and CD34^+^ MDS cells, and its level is correlated with apoptotic rates [121]. Overexpression of *TLR4* and *TLR9* were also reported and correlated with TNFα level but decrease during leukemic transformation [122]. Multiple mediators activate downstream TLR signaling which induces hematopoiesis and hematopoietic stem cell growth deregulation, but specific effects are not clear [115]. However, TLR activation induces differentiation of hematopoietic progenitors into monocytes/macrophages and DCs through MyD88-dependant and growth factor-independent pathways, particularly TLR7 and TLR8 [115,123]. Furthermore, NF-κB deregulation may enhance the differentiation and proliferative abnormalities characteristics of MDS by inducing the expression of pro-inflammatory cytokines and pro-survival factors [115,124]. In summary, dysregulation of several pathways in MDS involves Myddosome, Trifosome, Inflammasome, Necroptosome, and TGF-β pathways [125].

Even though there are a few descriptions of pDCs in MDS, circulating pDC precursors were described long ago in flow cytometry with the CD123^+^HLA-DR^+^ phenotype, as well as cDC precursors [126]. Few pDC and cDC precursors were observed in MDS in comparison to a healthy donor, but abnormal karyotypically patterns were observed in pDC and cDC precursors, suggesting a clonal evolution. The overall activation of cDC2s in MDS was observed, but pDCs and cDC2 presented a hypo-responsiveness to TLR-mediated stimulation, suggesting a chronic stimulation [127]. Concerning immune checkpoint expression, cDC1 in MDS from BM showed a high upregulation of ILT2 and a downregulation of TIM-3 [128]. Recently, two studies have provided updated knowledge about pDCs in MDS [100,129]. In the first one, DC subset proportions were lower in MDS BM than in healthy donor BM [129]. Only the pDC rate was increased in MDS BM in comparison to the healthy donor, particularly in MDS, with low blast count. In opposition, cDC1 and cDC2 rates decreased with the blast count increase. Transcriptome analysis highlighted a diminished capacity for sensing pathogen/damage associated with molecular patterns (PAMP/DAMP) by MDS-derived cell and involved *BTK*, *CARD9*, *IRAK4*, *IRF3/7*, *MyD88* and *SYK* downstream genes. Proliferation of CD4^+^ and CD8^+^ T-cells was reduced for all MDS subsets, which might suggest a negative interaction with pDC by tolerogenic functions. In opposition, in the second study, low pDC rate was associated with an inferior outcome for low- and high-grade MDS [100], suggesting an immunogenic function of pDCs. This reflects the complex balance between the tolerogenic and immunogenic functions of pDCs, with antagonistic roles according to the importance of the pDC infiltrate. Sorted pDC showed a mixture of neoplastic and normal cells by fluorescence in situ hybridization, suggesting clonal evolution but also a differentiation blockade from myeloid blast to pDCs [100]. In addition, ex vivo culture of pDCs induced a limited rate of pDCs. Thus, pDCs in MDS are a marker of leukemic stem cells dysfunctions, leading to pDCs maturation and activation responsible for immune response, whereas their low proliferative rate is a predictor of the worst outcome, suggesting immune cell dysfunction. Inflammatory dysregulation may reflect a hematopoietic niche dysfunction, leading to a chronic inflammatory response through aberrant cytokine production and to a tolerogenic response or exhaustion.

### 5.3. Mature pDC Proliferation in AML

Acute myeloid leukemia results from a combination of molecular events in hematopoietic stem cells (HSC) that block differentiation and drive proliferation [130]. Since the French–American–British (FAB) classification, which is based on morphology, cytochemistry, and dysmyelopoiesis observations, several updates have included cytogenetic and molecular abnormalities in AML classifications [2,114], which were correlated with three subsets of risk stratification in the last European Leukemia Network (ELN) score [131]. New therapeutic approaches regarding personalized medicine are under investigation based on AML biomarkers or molecular abnormalities [132]. Recently, two teams have better characterized MPDCP in AML under the pDC-AML denomination [3,4].

In the initial description, pDC-AML represents 5% of AML, and pDC cut off was established at 2% in BM [3,4]. pDCs proportion was greater in pDC-AML (mean 7.7%) than in AML without pDC excess (mean 0.03%) and healthy donor (mean 0.29%) [3]. Phenotypic studies have highlighted that pDCs present mature pDC markers CD123^+^ HLA-DR^+^ CD4^+^ CD303^+^ CD304^+^, but a lack of CD56 unlike BPDCN, and express aberrant markers such as myeloid markers (CD34^+^ CD13^+^) and lymphoid markers (CD5^+^ CD7^+^ CD22^+^ TdT^+^) [3,4]. CD34 expression reflects an immature pre-pDC stage [133]. Concerning CD123 expression level, it was greater on pDC than on blasts (CD123^+^ or CD123^−^). Furthermore, the CD123 expression level was lower on BPDCN than on pDC from pDC-AML and healthy donors, and lower on normal CD34^+^ progenitors than on blasts from pDC-AML and AML without pDC excess [3]. In addition, cTCL-1 expression level was lower on pDC from pDC-AML than BPDCN [4].

In pDC-AML, blasts were undifferentiated (FAB M0) in 73% of pDC-AML from the French cohort [4], but a monocytic presentation (FAB M4/M5) was observed in half pDC-AML in the American cohort [3,4]. M0-pDC-AML were mostly de novo *RUNX1^mut^* AML, whereas M4/M5-pDC-AML were secondary to MDS/MPN *RUNX1^mut^* AML. Thus, two subsets of pDC-AML are suggested: M0-pDC-AML and secondary monocytic pDC-AML, with specific mutation profiles [4]. Monocytes, pDC and blast carried the same mutations but not T-cells. *RUNX1* mutations were observed in 70% of pDC-AML [3] and 100% of M0-pDC-AML [4]. M0-pDC-AML were mostly associated with chromosome 13 gain, and *ASXL1*, *DNMT3A*, *SRSF2* and *SF3B1* (Figure 1) [4]. Moreover, RNA sequencing assay in pDC-AML highlighted the upregulation of genes involved in the IFN-driven pDC transcriptional program, particularly *IRF7*, *MX1* and *IFI35* in pDC from pDC-AML in comparison to normal pDC and blasts [3].

Furthermore, ex vivo culture highlighted that CD34^+^ blasts from pDC-AML were able to differentiate in pDC, with a greater rate of pDC obtained with pDC-AML blasts than CD34^+^ cord blood progenitors and blasts from *RUNX1^mut^* AML without pDC excess. No pDC differentiation was observed with *RUNX1^wt^* AML without pDC excess [3]. Those preliminary data suggest a maturation continuity involving a pre-pDC stage and a common progenitor between blasts, pDC and monocytes, as recently suggested [101,102].

## 6. Current Therapies and Perspective in pDC-AML

### 6.1. Current Therapies

Epidemiologic data highlighted predominant skin involvement in pDC-AML compared to AML without pDC excess, a greater proportion of adverse ELN risk, and poorer outcomes, including lower induction therapy rates, a greater relapse rate, and a lower overall survival median at 18.8 months vs. 36.7 months [3]. Adverse outcomes suggest a tolerogenic microenvironment, as described in solid cancers, especially since allogenic stem cell transplantation is described to increase adverse risk in *RUNX1^mut^* AML and pDC-AML [3,134]. This also suggests that allogenic stem cell transplantation is required to improve pDC-AML outcome.

On the other hand, L-asparaginase anti-leukemic affect is extremely well known in acute lymphoblastic leukemia [135]. It is also used in BPDCN treatment [58] and seems to be effective on AML leukemic stem cells [136]. Including L-asparaginase could be an interesting way to optimize chemotherapy regimens in pDC-AML, but further investigations are necessary to confirm its relevance.

### 6.2. Targeted pDC from pDC-AML

Recently, an anti-pDC strategy was investigated in pDC-AML, using the anti-CD123-directed diphtheria toxin tagraxofusp-erzs, which was able to eliminate the pDC population in pDC-AML. However, anti-leukemic activity was limited with only a reduction in the blast rate with tagraxofusp-erzs [3]. Other anti-CD123 therapies, now investigated in BPDCN [68] and AML [132], could similarly improve outcomes in pDC-AML. Moreover, using CD123 chimeric antigen receptor (CAR) T-cells would permit to target pDC and blasts in pDC-AML more efficiently, as reported in new preclinical and clinical approaches in BPDCN [137] and AML [138]. Finally, FLT3 bispecific T-cell engager (BiTE), reported in AML treatment [139] could be interesting in pDC-AML to target pDC, which expresses FLT3.

Immunotherapy is another strategy to modulate the tumor microenvironment and interactions between pDCs and T-cells. In AML and myelodysplastic syndromes, IDO1 expression has been identified as an independent adverse prognostic factor [111], able to impair immune response by Treg induction [140]. Using IDO1 inhibitor is promising in AML [141] and may be interesting in pDC-AML. T-cells from the AML BM microenvironment also express several immune checkpoint molecules such as PD1, CTLA-4, LAG3, TIM3, GITR, OX40, 4-1BB and ICOS, whereas AML blasts express 4-1BBL, CD80, CD86, ICOSL, PD-L1, PD-L2 and OX40L. Furthermore, those markers are correlated to *TET2*, *IDH1*, *IDH2,* and *FLT3* mutations [142].

Treg proportion in AML BM is also higher than in healthy donors. The expression of ICOSL by AML blasts promotes Treg accumulation in BM and blockade of ICOS signaling impaired Treg generation and retarded AML progression [143]. Epigenetic therapies such as hypomethylating agents upregulate PD-L1, PD-L2, PD-1, and CTLA-4 in MDS, CMML, and AML [106]. The V domain immunoglobulin suppressor of T cell activation (VISTA) was reported to be highly expressed on myeloid-derived suppressor cells and could inhibit CD8^+^ T-cell activity in AML, which expressed PD-1 [144]. The GITR blockade may enhance TAA-specific T-cells [85]. ASXL1 loss and *RAS* mutations were associated with T-cell expression of PD-1 and TIGIT [112]. The particular TIGIT^+^CD73^−^TOX^+^TCF-1^low^CD8^+^ T-cells phenotype was increased in AML, and TIGIT blockade could improve T-cell cytotoxicity [145]. CD47 blockade in AML was also investigated with interesting phagocytosis restauration, and AML and MDS eliminations [146]. Current clinical trials targeting PD-1, PD-L1 and CTLA-4 are now predominant and, in combination with an hypomethylating agent, aim to enhance both activities [147,148]. Of note, pDCs could also persist in the BM microenvironment even after anti-PD-1 therapies [149]. In addition, the use of vaccination in AML may overcome the limitations of the immune checkpoint blockade by evoking a clonal T-cell response [150]. All those immunologic strategies, investigated in AML, may take into account this immune pDC contingent to differentiate responses in AML with or without pDCs, and to optimize pDC-AML treatments through a novel immune checkpoint blockade in association with the conventional treatment actually used (Table 2).

Optimization of cancer immunotherapy could implicate DC modulations or targeting [91]. Moreover, a better characterization of DC lineages is important for a better understanding of DC functions in cancer, as well as in AML. Indeed, pDCs are able to present TAA, which could lead to an immune response, but the role of clonal pDC proliferation in immune response is still unknown [151]. However, TAA is derived mainly from supposedly non-coding regions (86%) in AML [151]. TAA-specific CD8^+^ T-cells reinjection strategies used in cancers have been optimized by transgenic T-cell receptor T-cells engineering (tgTCR T-cells), which enables us to recognize TAA specifically presented by selected MHC molecules [152,153,154]. Several studies are currently investigating different tgTCR T-cells constructions [132], but further characterizations of the pDC-AML entity are necessary in order to determine specific TAAs.

### 6.3. Targeted RUNX1 Mutations and pDC

The *RUNX1* mutation rate was higher in pDC-AML than in all AML, but *RUNX1* mutations were more frequently reported in de novo AML and M0-AML in comparison with all AML and pDC-AML [155,156,157,158]. In M0-AML, there is often a biallelic or a germline *RUNX1* mutation [158,159]. Furthermore, the same mutational profile was reported in *RUNX1^mut^* AML and *RUNX1^mut^* pDC-AML, with *ASXL1*, *DNMT3A* and *RAS* pathway mutations [159]. However, the *RUNX1* mutation rate in pDC-CMML was not increased and only concerned 3–12% of pDC-CMML versus 2–20% of pDC-poor CMML [99]. Thus, de novo *RUNX1^mut^* M0-pDC-AML seemed to have a particular clonal evolution with a pDC proliferation driven by *RUNX1* mutation, contrary to pDC-CMML.

On the other hand, *RUNX1* mutations are mainly a loss of function [3,158,160]. However, the regulation of the expression level between the RUNX1/2/3 transcription factors was reported [161], leading to the overexpression of RUNX2 in case of RUNX1 loss, which promoted pDC differentiation, *IRF7* expression and increased antigen presentation and ZEB 1 and 2 expression loss [19,161,162]. Moreover, RUNX1 loss of function induced dysregulated innate immune signaling [163]. MYD88-NF-κB signaling activated by TLR and IRAK4/TRAF6 recruitment led to NLR protein 3 (NLRP3) activation, adaptor apoptosis-associated speck-like protein containing a caspase recruitment domain (ASC) and pro-caspase-1 recruitment, and NLRP3-inflammasome formation, leading to caspase-1 activation and inflammatory cell death such as pyroptosis and necroptosis [164,165]. Dysregulation of innate immune signaling by RUNX1 loss originates in myeloid precursors, and RUNX1 regulates TLR signaling and inflammatory cytokines production [166]. In addition, loss of RUNX1 in CD4^+^ T-cell and Treg can cause inflammatory disorders [167,168]. RUNX1 loss also reduces ribosome biogenesis and generates cellular stress and an inflammatory microenvironment [169,170].

Targeting RUNX family abnormalities could be an interesting strategy in pDC-AML. Indeed, BET induces a TCF4 amplification loop [27]. The use of BETi was proposed in AML [132,171,172,173,174], BPDCN [61] and other cancers [175]. BETi was found to be efficient in *RUNX1^mut^* AML [174,176], and its association with other drugs induced synergic blasts lethality, notably with the JAK inhibitor ruxolitinib [171] and the anti-BCL2 agent venetoclax [176]. Furthermore, combined inhibition of MEK and BET was described to induce the downregulation of FLT3 and AP-1 expression to partially restore the immune microenvironment and to enhance CD8^+^ T-cells cytotoxicity [112]. Further clinical studies are required in the field of pDC-AML.

Finally, the regulation of pDC by fatty acid and lipid metabolism was also reported [177]. Indeed, inhibition of fat acid synthesis (using C75 and TOFA) blocking the Cpt1a function (using etomoxir or short-hairpin RNA) seemed to decrease TLR-induced production of IFNα, TNFα, and IL-6 by pDCs [178,179]. This could be another interesting new preclinical model [180].

## 7. Conclusions

A better understanding of pDCs biology, oncogenesis and immune response have permitted us to identify pDCs as a potent target in myeloid neoplasms. Through their tolerogenic activity, pDC are able to modulate innate and adaptive immune responses in myeloid neoplasm, responsible for adverse outcomes. Systematic research of pDC infiltration and pDC markers in BM at diagnosis or at relapse of myeloid neoplasm could predict adverse outcomes and may allow more appropriate treatments to be found. Regarding precision medicine, treatment targeting pDCs could be an interesting way to optimize outcomes observed in MPDCP associated with myeloid neoplasms, but there is still a need for further investigations. In the future, adoptive therapies such as CAR T-cells, tgTCR T-cells, as well as BiTE and other new targeting strategies and immunotherapies might be used as adjuvant treatments in the early stages of the disease.

## Figures and Tables

**Figure 1 cancers-14-03545-f001:**
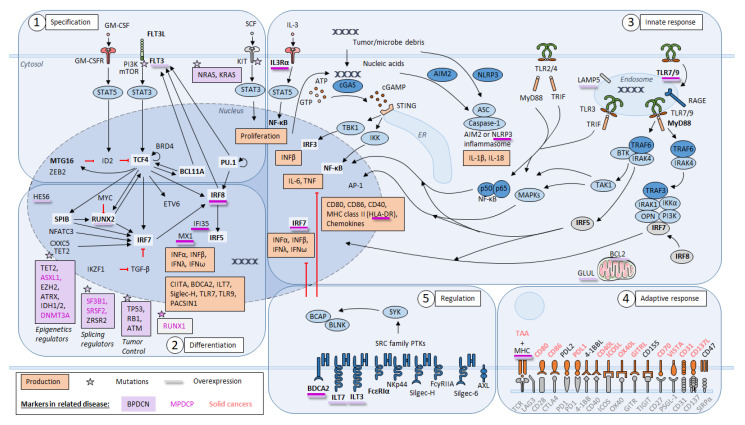
Plasmacytoid dendritic cell (pDC) development and functions. During development and activation of pDC, several networks are successively involved as specification program, differentiation program, innate and adaptive response pathways, and regulatory pathways. Specification program of pDC start in progenitors through PU.1 expression, FLT3 and TCF4 pathway leading to IRF8 activation. Differentiation into pDC depends to SPIB expression leading to IRF7, IRF8 and IRF5 activation, IFN-I (IFNα, IFNβ, IFNω) and IFN-III (IFNλ) secretion, and expression of several genes involved in pDC functions. Innate response develops from pattern recognition receptors (PPRs) such as Toll-like receptors (TLR), NOD-like receptors (NLRs), cytosolic DNA sensors (CDCs) and receptor for advanced glycation end products (RAGE) leading to IFN-I secretion, cytokines and chemokines secretion, and proteins expression involved in adaptive response. After activation and molecule of histocompatibility complex (MHC) class II antigen presentation, pDCs express several immune checkpoint molecules. Other regulatory molecules are expressed by pDC after IFN-I and cytokines production. Specific markers were reported in Blastic pDC Neoplasm (BPDCN, purple background), Mature pDC Proliferation (MPDCP, pink), and solid cancers (salmon red) including mutations (star) and overexpression (underline).

**Table 1 cancers-14-03545-t001:** Typical plasmacytoid dendritic cells phenotype.

Markers	Blastic pDC from BPDCN	pDC from MPDCP	Normal Mature pDC
lin (CD3, CD19, CD14, CD16, cMPO)	Negative	Negative	Negative
CD45RA	Positive	Positive	Positive
CD123	High	High	High
HLA-DR	High	High	Positive
CD4	Positive	Positive	Positive
CD303	Intermediate	Positive	High
CD304	Intermediate	Positive	High
CD11c	Negative	Negative	Negative
CD56	Positive	Negative	Negative *
CD34	Negative	Positive	Negative
CD5	Positive/Negative	Positive/Negative	Negative *
CD7	Positive/Negative	Positive/Negative	Negative
CD22	Positive/Negative	Positive/Negative	Negative
TdT	Positive/Negative	Positive/Negative	Positive
cTCL-1	High	Intermediate	Negative
nTCF4	Positive	Positive	Positive

Classical expression of markers are depicted, but aberrant expression are possible. pDC: plasmacytoid dendritic cells, AML: acute myeloid leukemia, BPDCN: blastic pDC neoplasm. Positive markers in blue scale upon intensity (light: low/intermediate, dark: high) * Some markers are positive in small cell populations.

**Table 2 cancers-14-03545-t002:** Potential therapies in pDC-AML.

Therapeutic Class	Targets	Type of Therapy
Chemotherapy	Asparagine	L asparaginase
Adoptive therapies	pDC & blasts	ASCT
	CD123	mAb, CAR T-cells
	FLT3	BiTE
	TAA	tgTCR, vaccination
Immune therapies	PD-1	Immune checkpoint molecule blockade
	PD-L1	
	CTLA-4	
	GITR	
	ICOS	
	TIGIT	
	CD47	
Targeted therapies	BET	Inhibitors
	MEK	
	BCL2	
	JAK2	
	IDO1	

pDC-AML: acute myeloid leukemia with excess of plasmacytoid dendritic cells; ASCT: allogenic stem cell transplantation; mAb: monoclonal antibody; CAR T-cells: chimeric antigenic receptor T-cells; BiTE: bispecific T-cell engager; tgTCR T-cells: transgenic T-cell receptor T-cells; TAA: tumor-associated antigen.

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
