# Peer review of "Plasmacytoid Dendritic Cells, a Novel Target in Myeloid Neoplasms"

_cancers, 2022, doi:10.3390/cancers14143545_

Round 1
Reviewer 1 Report
The review by Roussel et al. is a well written and comprehensive review on the still poorly known field of pDC/pDC-related malignancies.
I Would suggest the authors to refer to IDO1 rather than IDO (at the moment it's the more accepted nomenclature).
Minor check spelling is also required (especially in the first part of the review- lines 1-133). In the figure legend of Table2, "potent" should be changed to "potential".
Line 578 "adoptive immune responses" should be adaptive.
I would also suggest the authors to expand the "conclusions" paragraph with a critical and personal perspective on the clinical and therapeutic significance of an in-depth knowledge of both the phenotype and functions of pDCs in the described malignancies.
Author Response
The authors are sincerely grateful to the reviewer for his valuable remarks, including correcting spelling errors and advising on the relevant paragraphs to highlight :
“IDO” has been replaced by “IDO1” in lines 231, 234, 239, 287, 288, 348, 350, 351, 353, 356, 357, 358, 388, 493, 495, and in Table 2.
“Potent” has been replaced by “Potential” in Table 2
“Adoptive” has been replaced by “Adaptive” in lines 284, 578, 579 and in table 2.
Conclusion has been expanded (lines 586-597)
Reviewer 2 Report
This review address a not very well established chapter in cancer, and, especially, hematological neoplasms. The authors have tried to pool together the existing information related to the topic in the literature, including their significant work. However, their review looks now like a cascade of information pooled from different sources without a clear integrating and logic. Moreover, it does not flow well.
There is a lot of information in their multiple sections which can overwhelm the reader since there are not well related and looks like an endless enumeration. Where is applicable, and the description is very important for the topic, I would strongly suggest graphical representations of the discussed pathways.
I noticed that critical novel pathways are not addressed here. E.g.: Since the innate sensing of nucleic acids is recognized as the main triggering factor for interferon production by pDCs, the cGAS-STING pathway should have been discussed, including the agonistic and antagonistic avenues for targeting this pathway derived by the immunologic context and comorbidities.
There is a lot of confusion from Table 1. While I understand the controversial literature of this delicate topic, classifying a cell population as negative or positive may lead to false positive/negative gating in flow cytometry. The authors should have presented these information as flow charts with pros and cons to make understand the reader the complexity of this investigation.
I also observed that a lot of important references in the manuscript came from reviews. While, this is acceptable, for key definitions and also for the originality of the review, the authors should cited more original works.
In terms of editing, I strongly recommend a professional English proofreading.
There are too many keywords defined in the abstract page. In general, not more than 5 words are accepted. Please clarify in the line 578 if pDC modulate “adoptive” or “adaptive” immunity.
The word “interestingly” is used extensively in the manuscript.
There are also few redundant words such “myeloid neoplasm oncogenesis” (line 576).
Finally, I do not see the overall hallmark/signature of the authors related to the topic. What are their perspectives about this important chapter of immunology/neoplastic blood disorders?
Author Response
We carefully noted the areas of improvement proposed by the reviewer, as detailed below:
1/ Sections have been summarized in figure 1
2/ Paragraph about PAMPs/DAMPs recognition and interferon gene activation have been rewritten (lines 163-192)
3/ Table 1 have been clarified
4/ Original references have been included (10, 13, 45, 46, 47, 65, 66, 67)
5/ Despite 10 keywords were authorized, we reduce to 5 keywords.
6/ Redundant words have been reword, and confusion between adoptive/adaptive were corrected.
7/ Conclusion have been improved (lines 586-597)
Reviewer 3 Report
This Review, on “Plasmacytoid Dendritic Cells, a novel target in Myeloid Neoplasms”, is well written and comprehensive on important topic of the disease of Plasmacytoid dendritic cells in myeloid nepplasms in relation to Blastic pDC neoplasm and Tumor-infiltrating pDCs in solid cancers and . Although this manuscript has some useful information for the readers of this journal, it has the following limitations:
Major)
Page 10, In 5.2. Mature pDCs proliferation in MDS) “Recently, two studies update knowledge about pDCs in MDS [111,112]. In the first one, increase of pDC rates in BM was associated with MDS low blast count, while cDC1 and cDC2 rates decreased with blast count increase [111]. Transcriptome analysis highlighted a diminished capacity for sensing pathogen/damage associated molecular patterns (PAMP/DAMP) by MDS-derived cell and involved BTK, CARD9, IRAK4, IRF3/7, MyD88 and SYK downstream genes. In addition, proliferation of CD4+ and CD8+ T-cells was reduced for all MDS subsets. In the second study, low pDC rate was associated with an inferior outcome for low- and high-grade MDS [112].”
Apparently, Mature pDCs proliferation in MDS is associated with poor outcome. That is converse to those of Mature pDCs proliferation in CMMoL and AML. It is better to discuss on this point.
Minors)
Page 1) “Blastic Plasmacytoid Dendritic Cells Neoplasm, the blastic counterpart of pDC is well described,” is better to be “Blastic Plasmacytoid Dendritic Cells Neoplasm, aggressive leukemia derived from pDCs is well described,” as on page 3.
Page 5) “2.3. Transcriptomic network of pDC development”
Description of this section is better to be reduced with/without a figure.
Page 6) “3. Blastic pDC neoplasm”
Cyto/histological findings should be described at least in brief.
Concerning the treatment of this neoplasm, a reference as shown in below is better to be added.
Pemmaraju N, Lane AA, Sweet KL, et al. Tagraxofusp in blastic plasmacytoid dendritic-cell neoplasm. N Engl J Med. 2019;380(17):1628–1637.
Page 7) “4. Tumor-infiltrating pDCs in solid cancers”
Description of this section is better to be reduced with/without a figure.
Page 8) “Origin of pDCs observed in hematological malignancies is unclear between tumor infiltrations or clonal maturation of pDCs from a primary hematological malignancy”
The sentence is unclear.
Page 8) “It was reported in association with CMML, AML or MDS, more recently referred to as pDC-CMML, pDC-AML and pDC-MDS respectively.”
The 3 references are better to be added here.
Author Response
The authors would like to thank the reviewer for his constructive feedback, notably to adjust some key points and to emphasize relevant paragraphs :
Major: Page 10): paragraph has been rewritten and clarifications have been made (lines 407-434)
Minor:
- Page 1): sentence has been reworded (lines 24 and 53)
- Page 5): part 2.3 has been reduced in figure 1
- Page 6): in part 3, cyto/histological findings has been added as published by Garnache et al in Blood Adv., 2019 (lines 196-199). Treatment references have been updated with Pemmaraju NEM 2019 reference addition (]67]), and sentence has been rewritten (lines 227-228).
- Page 7): part 4 has been reduced
- Page 8): “Origin of pDCs observed in hematological malignancies is unclear between tumor infiltrations or clonal maturation of pDCs from a primary hematological malignancy” has been deleted, and paragraph has been rewritten and references have been updated (lines 299-310)
Round 2
Reviewer 2 Report
I would like to congratulate the authors for their extensive reviewing and editing. I think that the manuscript in the present form is suitable for publication.
Reviewer 3 Report
I think the manuscript has been revised well.